# Effect of interstitial palladium on plasmon-driven charge transfer in nanoparticle dimers

Sarah Lerch[1] & Björn M. Reinhard [1]

Capacitive plasmon coupling between noble metal nanoparticles (NPs) is characterized by an increasing red-shift of the bonding dipolar plasmon mode (BDP) in the classical electromagnetic coupling regime. This model breaks down at short separations where plasmon-driven charge transfer induces a gap current between the NPs with a magnitude and separation dependence that can be modulated if molecules are present in the gap. Here, we use gap contained DNA as a scaffold for the growth of palladium (Pd) NPs in the gap between two gold NPs and investigate the effect of increasing Pd NP concentration on the BDP mode. Consistent with enhanced plasmon-driven charge transfer, the integration of discrete Pd NPs depolarizes the capacitive BDP mode over longer interparticle separations than is possible in only DNA-linked Au NPs. High Pd NP densities in the gap increases the gap conductance and induces the transition from capacitive to conductive coupling.

---

[1] Department of Chemistry and The Photonics Center, Boston University, 8 Saint Mary's Street, Boston, MA 02215, USA. Correspondence and requests for materials should be addressed to B.M.R. (email: bmr@bu.edu)

Plasmons are coherent oscillations of the conduction band electrons in metals or doped semiconductors, and noble metal nanoparticles (NPs) sustain localized surface plasmon resonances in the visible range of the electromagnetic spectrum[1,2]. The plasmons of individual NPs can couple in the near-field if two or more NPs are separated by less than approximately one NP diameter[3,4]. In the classical electromagnetic coupling regime for NPs that are small compared to the resonance wavelength, the near- and far-field responses of a NP dimer will be dominated by the bonding dipolar plasmon (BDP) mode under optical excitation[5–9]. The BDP mode arises from the capacitive coupling of the plasmons localized on the two NPs, and this mode red-shifts relative to the single-particle resonance with decreasing interparticle separation[10–12]. The spectral shift is accompanied by a surface charge accumulation and increasing electric (E-) field localization in the gap region. This classical electromagnetic behavior breaks down at very short interparticle separations where quantum mechanical effects become relevant[13–16]. Electron transfer between the NPs reduces the charge build-up on the two coupled NPs, resulting in the stagnation or even blue-shift of the BDP mode compared to the classical electromagnetic prediction[17–20]. The depolarized blue-shifted BDP mode is also referred to as screened BDP[21,22]. For high gap conductances, an additional resonance, the so-called charge transfer plasmon (CTP), emerges in the infrared (IR) region of the electromagnetic spectrum[23,24]. The characterization of the role of the different plasmon modes, especially the CTP, in facilitating electron transfer across nanogaps in the quantum plasmonic regime, and the relative contributions from photoassisted transport and hot electron driven mechanisms, have been the subject of a series of experimental and theoretical studies[23–27]. In general, quantum effects become relevant in gap structures with separations of approximately 0.5 nm and below in vacuum, but this range can be extended if molecules are present in the gap[21,28–31]. Hybrid systems comprising metal nanostructures and connecting molecules are of particular relevance in materials science as the molecularly guided self-assembly of plasmonic NPs provides a rational and scalable fabrication strategy of plasmonic biosensors, nanoantennas, and components for complex plasmonic nanocircuitry[16,32–35]. The most common molecular linker in self-assembled plasmonic structures is DNA, as its four base coding structure allows for a highly selective binding interaction and programmable assembly via hybridization of complementary DNA strands[36–40]. The use of DNA as the linker between NPs led to dynamic molecular rulers, referred to as plasmon rulers (PRs)[41–43], and with the advent of DNA origami, the precision of DNA-templated NP assemblies has further improved[44]. Intriguingly, DNA is not necessarily an inert building block. Instead, DNA can mediate charge transfer through mechanisms that are highly distance dependent[45–51]. Coherent tunneling has been observed for separation up to approximately 3 nm[51–53], and charge transfer over longer separations is still feasible through sequential hopping events. The role of DNA-mediated conductance on the transition from classical electromagnetic to quantum mechanical coupling regime in DNA-connected NPs raises important fundamental questions. Although the classical electromagnetic coupling model, which predicts a continuous red-shift with decreasing gap separation, provides, in general, an adequate description of most practical DNA-linked NP assemblies in solution, correlated optical and high-resolution characterization on the single dimer level have provided indications that at short separations the DNA itself influences the coupling between gold (Au) and silver (Ag) NP dimers[30,43]. Significant blue-shifts of the BDP mode relative to the classical electromagnetic prediction for distances of up to $S_{thresh} = 2.8$ nm observed for immobilized dimers immersed in oil or glycerol were

attributed to DNA-mediated charge transfer between the gap defining NPs. Similar blue-shifts were observed—albeit at shorter separations—with other molecular linkers that provide gap conductance[28,29,34,54]. Intriguingly, DNA origami templated dimers with sub-5-nm interparticle separations, whose gap region was only partially filled with DNA strands oriented perpendicular to the interparticle axis[55], did not show signs of a weakening of the electromagnetic coupling. Together, these observations indicate a sensitive dependence of the optical response of self-assembled NP dimers at short interparticle separations on details of the gap configuration.

Motivated by the prospect that gap conductance is a control parameter that makes it possible to modify the distance-dependent plasmon coupling in self-assembled NP assemblies and the independent finding that the integration of a third NP between two strongly coupled gold NPs enhances the range of coherent energy transfer[56], we set out to investigate the effect of interstitial metal NPs on the distance-dependent plasmon coupling between DNA-connected Au NPs. Metallization by templated growth of small (2–5 nm diameters) NPs is a common strategy for improving the conductivity of DNA[57–60]. Here, we investigate the effect of increasing palladium (Pd) NP intragap density on the optical response of DNA-tethered 41 nm (±16 nm) Au NPs and map the transition from capacitive to conductive coupling. Pd is chosen as $Pd^{2+}$ cations are known to bind to DNA, and hierarchical Pd–Au heterostructures have interesting sensor applications. We characterize the effect of discrete Pd NPs localized in the gap on BDP depolarization in the capacitive coupling regime through correlated optical single dimer spectroscopy and high-resolution transmission electron microscopy (TEM) and test the hypothesis that the generation of Pd NP in the gap of DNA-connected plasmonic molecules provides a chemical strategy to tune their collective plasmon resonance after self-assembly[61].

## Results

**Incorporation and growth of Pd NPs in PR gaps**. DNA-linked Au NP dimers, referred to in the following as PRs[41–43], were assembled through DNA-programmed self-assembly outlined in Fig. 1a. Briefly, two aliquots of 41 (±16) nm Au NPs were incubated with complementary single-stranded DNA (ssDNA) handles, followed by incubation with an excess of carboxylic acid-terminated polyethylene glycol (PEG-COOH) to ensure colloidal stability. After removal of excess ssDNA and PEG-COOH through centrifugation and resuspension, the two different NP flavors were combined and hybridized by incubation for several hours at 60 °C and subsequent annealing. We generated PRs with different handle DNA/NP ratios of 15/1, 22/1, and 30/1. After isolation of successfully formed dimers through gel electrophoresis (Supplementary Fig. 1)[62], Pd NPs were integrated into the DNA linker through a two-step DNA-templated reduction procedure (Fig. 1b). In the first step, $Pd^{2+}$ was bound to DNA. Subsequently, the DNA-bound $Pd^{2+}$ ions were reduced with 4 mM dimethylamine borane (DMAB), resulting in the coalescence of Pd NPs. Our experimental strategy was adapted from ref. [55] and is described in detail in the Methods section. Successful $Pd^{2+}$ binding to the DNA was confirmed through the red-shift of the plasmon resonance wavelength distribution of the PR, shown in the insets in Fig. 1b. The red-shift results from an increase in the local refractive index associated with the binding of $Pd^{2+}$ ions to DNA. Additional confirmation of $Pd^{2+}$ binding is given by the neutralization of the zeta potential of the PRs (Supplementary Table 1). Figure 1c–e shows TEM images of PEG-COOH-functionalized NPs (no DNA, Fig. 1c) and PRs with DNA/NP ratios of 15/1 (Fig. 1d) and 30/1 (Fig. 1e). PEGylated NPs had no

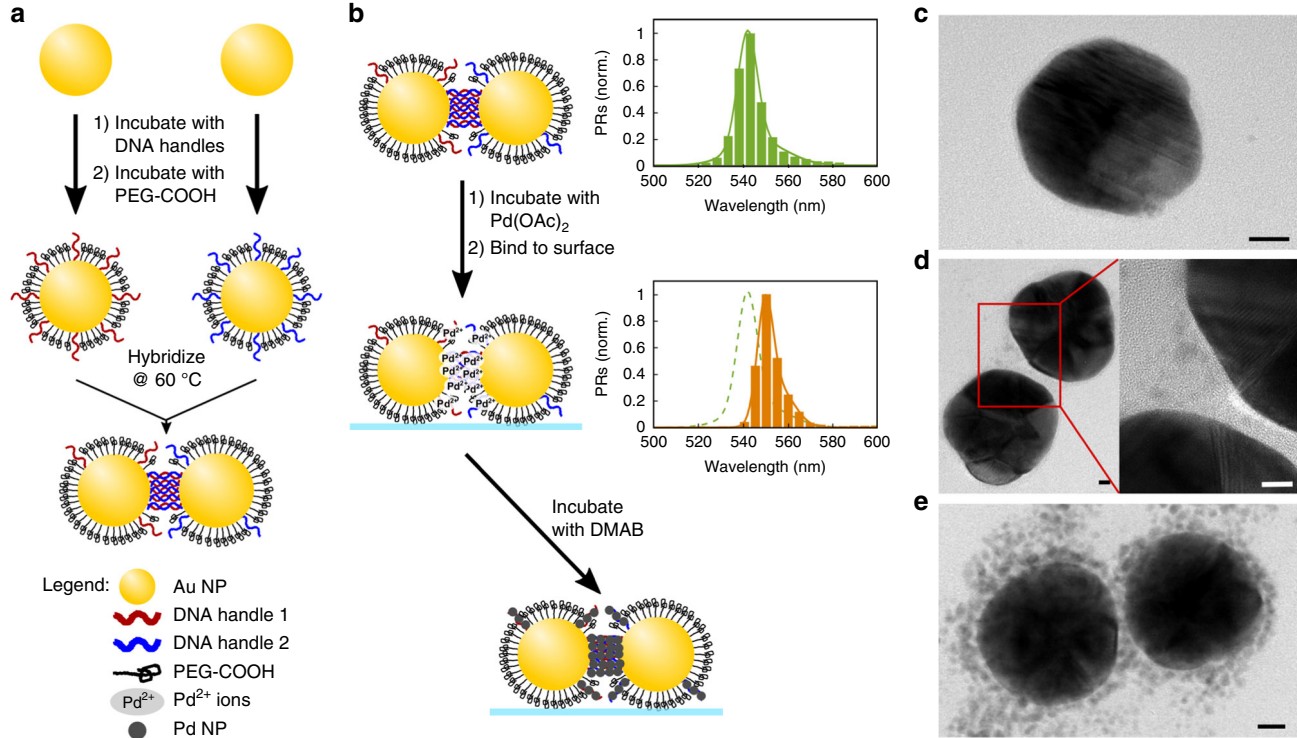

**Fig. 1** DNA-guided growth of Pd NPs in plasmon rulers. **a** Model for plasmon ruler (PR) assembly from Au NPs functionalized with a low number of DNA molecules incorporated into a monolayer of PEG-COOH for colloidal stability. **b** Integration of Pd NPs. $Pd^{2+}$ binds to DNA and is subsequently reduced through the reducing agent DMAB. Distributions of the plasmon resonance wavelength of PRs before and after $Pd^{2+}$ binding are included as green and orange histograms, respectively. **c** TEM image of PEGylated NP control (scale bar = 10 nm). **d** TEM image of PR with low Pd NP loading after reduction of $Pd^{2+}$. (scale bars = 5 nm). **e** TEM image of PR with high Pd NP loading after reduction of $Pd^{2+}$ (scale bar = 10 nm)

or only a very low number of Pd NPs attached. For PRs, the number of Pd NPs showed some variability, but the number and average size of Pd NPs bound to the DNA-functionalized NP increased overall with DNA loading (Supplementary Table 2). For low DNA handle concentrations, some PR contained Pd NP exclusively localized to the gap region, which we attribute to a residual lateral mobility of DNA handles on the NP surface[63]. DNA handles can diffuse into the gap region where they get trapped upon hybridization with a complementary strand bound to the second NP. Overall, this process results in an enrichment of DNA in the gap region.

**Darkfield analysis of the reduction of $Pd^{2+}$ to Pd NPs.** In the next step, we evaluated the spectral changes associated with Pd NP formation on the optical response of single 15/1 DNA/NP PRs in a massively parallel fashion in real time. PRs were pre-incubated with $Pd^{2+}$, washed, and then immobilized on the inner surface of a poly-L-lysine-treated flow chamber that was imaged with a darkfield microscope at 60× magnification. A typical field of view with dimensions up to 143 by 143 $\mu m^2$ contained approximately 500 individual scatterers (Supplementary Fig. 2), of which typically 300–400 were PRs. The majority of NP monomers were excluded from all subsequent analyses through an intensity thresholding, and our analysis focuses primarily on dimers. A small fraction of monomers was, however, retained due to some overlap between monomer and weakly coupled PR intensity distributions. We first applied a ratiometric imaging approach to detect spectral changes triggered by the formation of Pd NPs upon addition of DMAB reducing agent in real time. We split the detected light using a dichroic mirror (580 nm) and monochromatic wavelength filters (550 nm and 600 nm) and

recorded two wavelength images on two separate cameras, shown in the schematic in Fig. 2a. The 550 nm channel coincides with the initial PR resonance and the 600 nm channel lies on the red tail of the PR spectrum. The comparison of the relative intensities of the two wavelength channels makes it possible to detect shifts of the plasmon resonance wavelength in real time. Figure 2b–d shows representative trajectories of controls (PRs without $Pd^{2+}$ pre-treatment but incubated with DMAB) and of Au PRs during $Pd^{2+}$ reduction. The trajectories were recorded with a rate of 1 frame per second and the reducing agent (4 mM DMAB) was flushed into the chamber with a rate of 0.8 $\mu L/s$.

While the PR control (no $Pd^{2+}$) in Fig. 2b shows no significant change in the relative intensities of the 550 nm and 600 nm channels, upon addition of the reducing agent, formation of Pd NPs induces measurable spectral shifts. Intriguingly, both red-shifts (Fig. 2c) and blue-shifts (Fig. 2d) are observed for PRs pre-incubated with $Pd^{2+}$ after addition of the reducing agent. To determine the frequency of blue- and red-shifts and quantify their magnitudes, we measured the spectra of hundreds of individual 15/1 and 30/1 DNA/NP PRs in the field of view through hyperspectral imaging in a separate widefield darkfield microscope (Fig. 3a). We recorded 21 monochromatic images in steps of 10 nm in the spectral range between 500 nm and 700 nm before and after the reduction of $Pd^{2+}$ cations to form Pd NPs using a tunable filter. The measured intensities of the individual PRs in the monochromatic images were then used to construct their scattering spectra, and the longitudinal BDP resonance wavelengths were determined by Gaussian fits around the peak intensity. We verified the accuracy of the spectral information obtained through hyperspectral imaging for a random group of PR using a conventional imaging spectrometer (Supplementary Fig. 3).

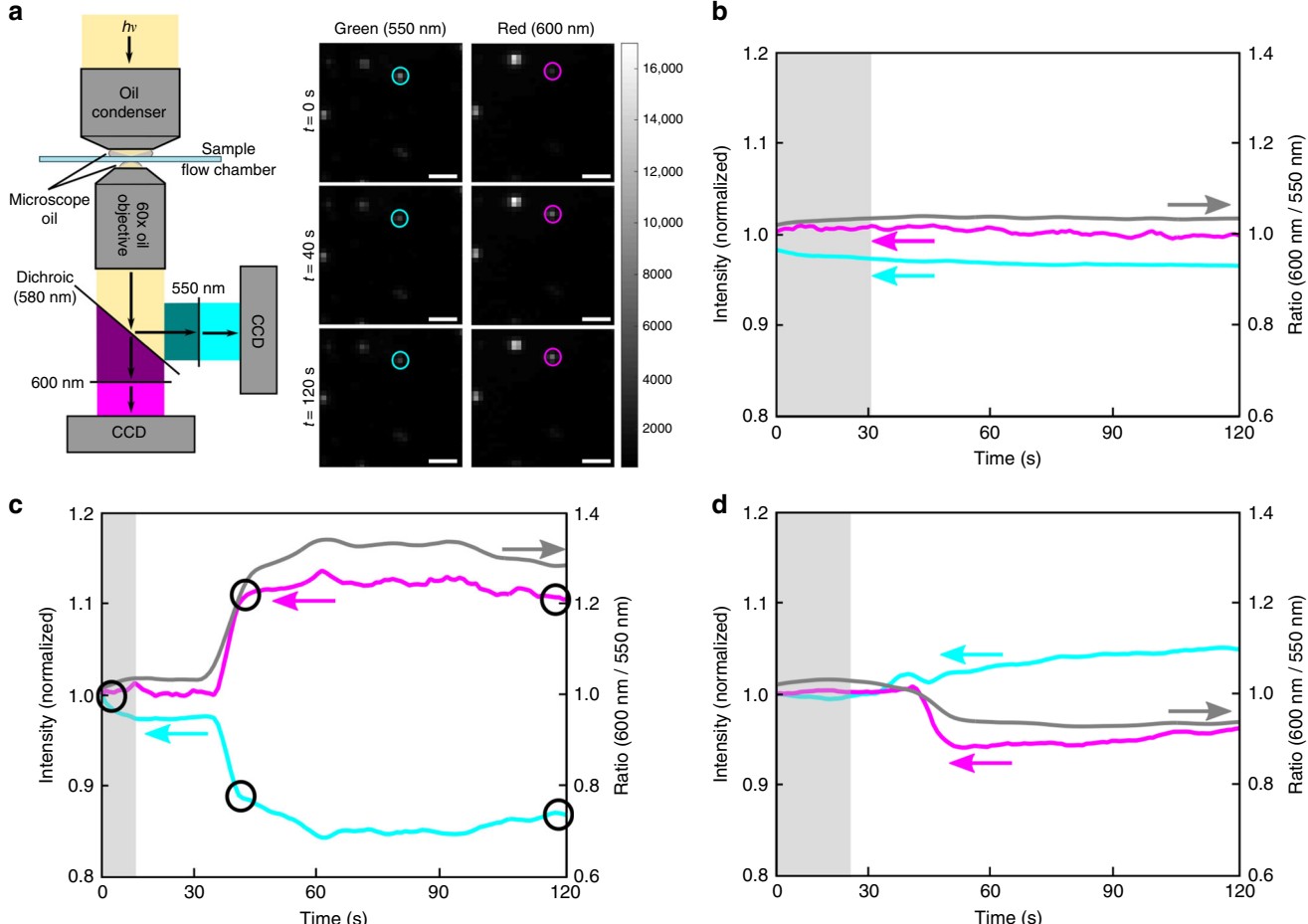

**Fig. 2** Detection of resonance shifts during the formation of Pd NPs. **a** Schematic drawing of the ratiometric imaging set-up, with representative CCD images on the monitored wavelength channels of 550 and 600 nm (magnified sub-section of the entire field of view, scale bar = 10 μm) at time points $t = 0$, 40, and 120 s. Plots of the monitored intensities as function of time for the circled point are shown in (**c**). **b** Intensities and calculated intensity ratios of PR controls (no Pd²⁺) during the incubation with DMAB. Gray line represents the ratio of the red (600 nm, red line) and green (550 nm, green line) wavelength channels. The shaded area marks an interval without flow before the reducing agent was flushed in. **c, d** Sample trajectories for 15 DNA/NP PR during the reduction of Pd²⁺ to Pd NPs showing **c** a red-shift and **d** a blue-shift

Figure 3b correlates the resonance wavelengths of PRs with 15 DNA/NP measured before ($\lambda_{initial}$) and after ($\lambda_{final}$) Pd²⁺ reduction. The resonance wavelengths of controls (PRs treated with DMAB in the absence of Pd²⁺) and individual NPs (monomers) in Pd²⁺ containing aqueous solution are included for comparison. The $\lambda_{initial}$ distribution of the PRs is red-shifted relative to that of the monomers due to near-field coupling between the NPs. The broad spread of the PR $\lambda_{initial}$ values reflects a large variability in the interparticle separation of the immobilized PRs. Importantly, different from the PR control for which $\lambda_{final}$ and $\lambda_{initial}$ are strongly correlated and the majority of the population (92%) shows a spectral shift, $\Delta\lambda = \lambda_{final} - \lambda_{initial}$, of less than ±10 nm (black line), the Pd²⁺ containing PRs show a large distribution of positive and negative $\Delta\lambda$ after addition of DMAB. Only 27% of PRs cluster around the $\Delta\lambda = 0$ (±10 nm) line, while 43% of PRs experience a red-shift of $\Delta\lambda > +10$ nm and, intriguingly, 30% of PRs show a blue-shift of $|\Delta\lambda| > |-10|$ nm as a consequence of Pd NP formation. Figure 3c shows the spectra before and after Pd²⁺ reduction of a PR that experiences a strong blue-shift of $\Delta\lambda = -95$ nm from $\lambda_{initial} = 645$ nm to $\lambda_{final} = 550$ nm. In principle, the conversion of a dimer into a monomer through loss of a NP could account for a spectral blue-shift. However, due to the absence of a systematic blue-shift in the PR (no Pd²⁺) controls and PR scattering intensities that remain significantly higher than those

of monomers (Supplementary Fig. 4), we exclude loss of a NP as a trivial reason for the observed blue-shift.

In addition to the induction of spectral shifts, the integration of Pd NPs also affects the width of the individual PR spectra. In the absence of Pd²⁺, PRs assembled from NPs with 15 DNA/NP have an average full width at half maximum (FWHM) of 78 ± 12 nm, which increases to 98 ± 31 nm after binding of Pd²⁺. The creation of Pd NPs through reduction of Pd²⁺ in the next step further broadens the single PR spectra. The 75% of the PRs that exhibit a blue-shift upon Pd NP integration show a significant broadening. The average FWHM increases from 103 nm ± 24 nm for Pd²⁺ loaded PR to 139 nm ± 34 nm after reduction of the Pd²⁺ ions. Similarly, 80% of the red-shifting sub-population spectrally broaden. The FWHM before and after Pd²⁺ reduction in this case are 100 nm ± 31 nm and 144 nm ± 31 nm, respectively. The observed spectral broadening shows that the metallic Pd NPs located in the near-field of the gold NPs represent parasitic elements that dampen the coupled plasmons of the PRs. The dampening results from dissipative losses in the small Pd NPs as well as from a heterogeneous modulation of the carrier density (and dielectric function) across the PR after introduction of a new metallic component.

The spectral shifts, $\Delta\lambda$, upon formation of Pd NPs measured for individual PRs assembled from Au NPs with 30/1 DNA/NP

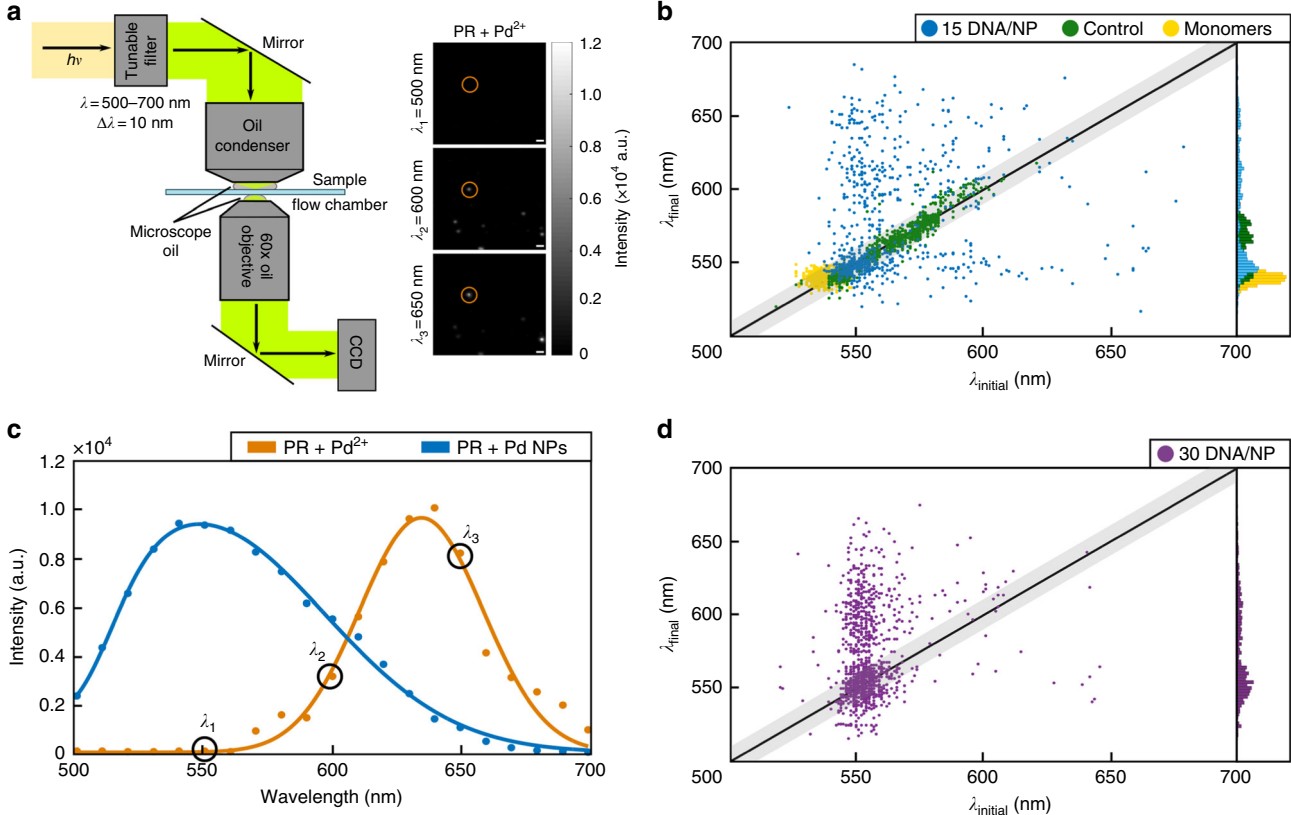

**Fig. 3** Hyperspectral imaging of $Pd^{2+}$ loaded PRs before and after reduction. **a** Schematic overview of hyperspectral darkfield imaging set-up with representation CCD images for $\lambda = 550$, 600 and 650 nm. Images are magnified sub-sections of a larger field of view (scale bar = 20 μm). The spectrum of the circled scatterer is shown in (**c**). **b** Plot of the plasmon resonance after $Pd^{2+}$ reduction ($\lambda_{final}$) as a function of resonance wavelength before $Pd^{2+}$ reduction ($\lambda_{initial}$) for 1031 individual PRs with 15 DNA/NP (blue). $\lambda_{initial}$ and $\lambda_{final}$ for control PRs (identical treatment but without $Pd^{2+}$) and NP monomers are included in green and yellow, respectively. The inset contains the distribution of $\lambda_{final}$. **c** Exemplary hyperspectral data for an individual PR with 15 DNA/NP whose plasmon resonance blue-shifts upon Pd NP formation. **d** Plot of the plasmon resonance after $Pd^{2+}$ reduction ($\lambda_{final}$) as function of resonance wavelength before $Pd^{2+}$ reduction ($\lambda_{initial}$) for 1056 individual PRs with 30 DNA/NP

are summarized in Fig. 3d. Intriguingly, the fraction of PRs that experience a blue-shift has dropped to 12%, while the fraction that experiences a red-shift has increased to 55% in this case. The 15/1 and 30/1 DNA/NP differ in the concentration, size, and spatial arrangement of the formed Pd NPs (see Fig. 1d, e). The significant decrease of the blue-shifted PR sub-population for PRs assembled from NPs with 30/1 DNA/NP that contain more and larger Pd NPs is a first indication that the blue-shift is favored by a lower concentration of Pd NPs in the gap region.

**Correlative darkfield, TEM, and electromagnetic simulations**. To understand the effect of Pd NP formation on the PR spectra in more detail and to elucidate the structural differences that result in spectral blue- or red-shifts, we combined in the next step single PR spectroscopy with high-resolution TEM. This strategy allowed the correlation of single PR structures with the associated spectra. This correlative optical/electron microscopy procedure has been described in detail previously[30] and relies on pattern recognition, aided by 2 μm polystyrene beads, to effectively align the darkfield and TEM images. Consistent with our hyperspectral data from Fig. 3, we found PRs with a blue-shift ($\Delta\lambda < 0$), a red-shift ($\Delta\lambda > 0$), or no shift ($\Delta\lambda = 0$) relative to the classical electromagnetic prediction after Pd NP formation. An overview of 20 PR structures and correlated spectra for different Pd NP loadings and interparticle separations is provided in Supplementary Fig. 5 and 6, respectively. Figure 4 shows selected examples for PRs assembled from NPs with 15 DNA/NP with $\Delta\lambda < 0$ (Fig. 4a-c), $\Delta\lambda > 0$

(Fig. 4d), and $\Delta\lambda = 0$ (Fig. 4e) that illustrate the dependence of the spectra on interparticle separation, $S$, and Pd NP density in the gap. The left column in each row contains a TEM image of an individual dimer, the middle row contains the experimental scattering spectrum (blue), the finite-difference time-domain (FDTD) simulated spectrum of the dimer embedded in index matching glycerol with constant refractive index of $n = 1.570$ (including the gap region), and the corrected FDTD simulation[64,65] that accounts for a DNA-mediated static gap conductivity of $\sigma_0 = 7.8$ S/m as well as the presence of Pd NPs in the gap region (orange) if applicable. The right column shows the structural PR model used in the corrected FDTD simulation and a charge density map for the supported plasmon modes. In the following, we refer to the peak wavelength of the experimental scattering spectrum as $\lambda_{exp}$; and $\lambda_{sim}$ and $\lambda_{corr}$ correspond to the peak wavelengths of the FDTD simulated spectra with and without correcting for DNA and Pd NPs, respectively. We also added the effective gap conductance, $G$, of a homogenous medium filling the space between the two Au NPs that in FDTD simulations best reproduces the experimental spectra of the individual PR in the right column of Fig. 4.

Figure 4a shows a PR with a short interparticle separation of $S = 2$ nm. Intriguingly, the measured BDP resonance wavelength of $\lambda_{exp} = 586$ nm is much shorter than the FDTD prediction of $\lambda_{sim} = 651$ nm. The interparticle separation $S = 2$ nm is below the threshold separation $S_{thresh} = 2.8$ nm for which we have previously observed a characteristic blue-shift relative to the classical electromagnetic prediction to occur, and which we attributed to a

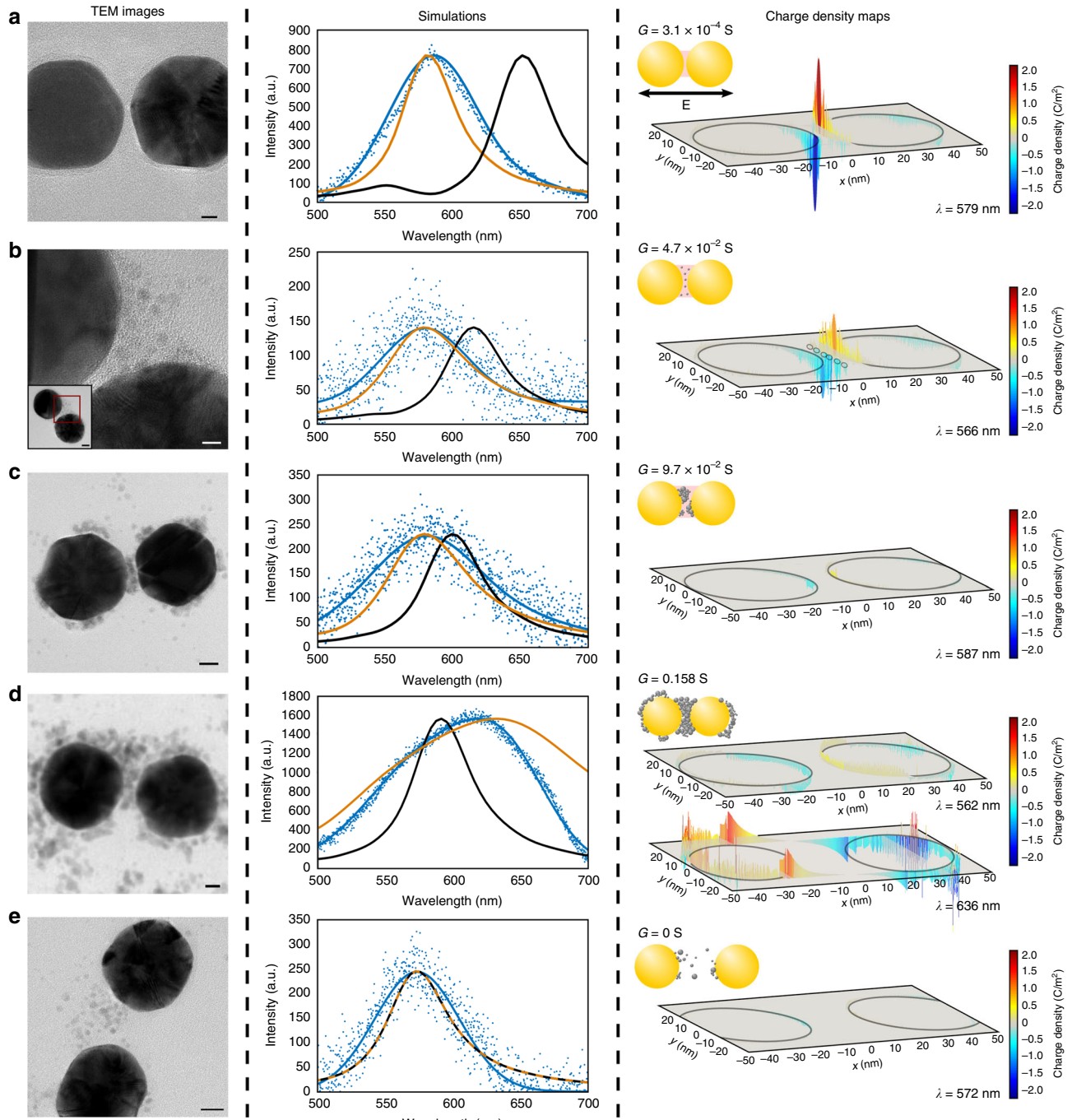

**Fig. 4** Correlated spectroscopy and TEM analysis of Pd NP containing PRs. **a** Gap width, $S = 2.0$ nm; experimental peak wavelength (blue), $\lambda_{exp} = 586$ nm; peak wavelength of the FDTD simulated spectrum (black), $\lambda_{sim} = 651$ nm; peak wavelength of the corrected FDTD simulations (orange), $\lambda_{corr} = 580$ nm. Scale bar = 5 nm. **b** $S = 4.0$ nm; $\lambda_{exp} = 575$ nm; $\lambda_{sim} = 612$ nm; $\lambda_{corr} = 566$ nm. Scale bar = 5 nm. **c** $S = 5.6$ nm; $\lambda_{exp} = 580$ nm; $\lambda_{sim} = 599$ nm; $\lambda_{corr} = 578$ nm. Scale bar = 10 nm. **d** $S = 8.3$ nm; $\lambda_{exp} = 621$ nm (shoulder $\lambda_{exp} = 569$ nm); $\lambda_{sim} = 590$ nm; $\lambda_{corr} = 632$ nm (shoulder $\lambda_{corr} = 562$ nm). Scale bar = 10 nm. **e** $S = 18.0$ nm; $\lambda_{exp} = 573$ nm; $\lambda_{sim} = 572$ nm; $\lambda_{corr} = 572$ nm. Scale bar = 10 nm

DNA-mediated gap conductance[30,43]. Indeed, the corrected FDTD simulations that account for DNA-mediated gap conductance yield a peak resonance wavelength of $\lambda_{corr} = 580$ nm and successfully reproduces the experimental spectrum for the dimer in Fig. 4a. The PR with short interparticle separation in Fig. 4a does not contain a Pd NP in the gap. For completeness we add that an increase in gap conductance due to an interstitial Pd NP that touches both Au NPs and forms a conductive bridge between the Au NPs provides a very similar spectrum (Supplementary Fig. 7). Consequently, at very short interparticle

separations a moderate gap conductance mediated either by DNA or discrete contacts established by Pd NPs accounts for a significant spectral blue-shift.

The PR in Fig. 4b has a longer gap separation of $S = 4.0$ nm, and the TEM shows individual Pd NPs interspersed in the gap. Although the Pd NPs do not establish a continuous metallic connection between the two Au NPs, and the interparticle separation is significantly longer than $S_{thresh}$, the BDP resonance is still remarkably blue-shifted ($\lambda_{exp} = 575$ nm) when compared to the classical electromagnetic prediction ($\lambda_{sim} = 615$ nm). $S$ is

too long to allow for direct tunneling from one Au NP to the other, but the separations between the individual Pd NPs fall below $S_{thresh}$. The measured BDP blue-shift indicates, thus, a non-negligible gap conductance that arises from multiple subsequent plasmon-driven and DNA-mediated tunneling events between Au and Pd NPs, as well as between Pd NPs in the gap. This model, which is reminiscent of electron percolation in films of NPs connected by organic molecules[66–72], is supported by the spectral blue-shift obtained in FDTD simulations that contain discrete 2 nm diameter Pd NPs integrated into a DNA matrix with a static conductivity of $\sigma_0 = 7.8$ S/m in the gap region. In excellent agreement with the experimental results, this model yields a peak resonance wavelength of $\lambda_{corr} = 566$ nm for a gap width of $S = 4$ nm. We observed this characteristic spectral blue-shift in Pd NP containing PRs with separations up to at least $S = 4.4$ nm (Supplementary Fig. 5j).

An example of third class of frequently observed structures is shown in Fig. 4c. PR of this type show significant accumulation of Pd NPs in the gap, but the Pd NPs do not yet establish a continuous connection between the Au NPs over large areas of the gap region. For the PR in Fig. 4c, the TEM image show a narrow gap with low electron density (contrast) between the Pd NPs and the surface of the right Au NP. The edge-to-edge separation between the two Au NPs is $S = 5.6$ nm, and the experimental spectrum of this PR peaks at $\lambda_{exp} = 580$ nm, which compares with $\lambda_{sim} = 599$ nm. We attribute the shift between $\lambda_{exp}$ and $\lambda_{sim}$ in this case to the fact that the Pd NP accumulation sufficiently narrows the gap to allow for some gap current between the Au NPs through (i) the formation of a limited number of point contact between the Pd NP cluster in the gap and the Au NP surfaces and/or (ii) direct tunneling across the remaining metal-free gap space. If the width of the metal-free region drops below $S_{thresh}$, DNA-mediated charge transport across the junction becomes feasible even in the absence of a direct contact. The relative spectral shift between $\lambda_{exp}$ and $\lambda_{sim}$ is weaker for Fig. 4c than for Fig. 4a, b due to the larger gap separation, which reduces the coupling between the particle plasmons. This is well illustrated by the charge density plots in the right column that show a progressive decrease in the gap region as the interpaticle gap is increased for Fig. 4a–c.

If the Pd NP concentration is further increased, the nature of the coupled plasmon dimer fundamentally changes. This is illustrated in Fig. 4d where the entire PR gap region is filled with inter-connected Pd NPs that also connect to both Au NP surfaces to establish a conductive connection across the entire PR. While for Fig. 4a–c, the coupling between the two Au NPs is capacitive in nature with a charge build-up on the opposite sides of the gap, for the structure modeled in Fig. 4d, two modes at $\lambda_{corr} = 562$ nm and $\lambda_{corr} = 636$ nm can be identified with the dominating longer wavelength mode showing clear indication of conductive coupling. In particular, charge accumulation occurs for this mode only on the outer surfaces of the Au NPs but not in the gap region, which is a reliable indicator of a transition from capacitive to conductive coupling due to a sufficiently high gap conductance. The mode at $\lambda_{corr} = 562$ nm could, in principle, represent a quadrupole mode, as these have been shown to couple to far-field radiation in dimers of touching NP[73]. The simulated charge distribution and the fact that the mode remains lower in energy than the BDP of the uncoupled monomer, whereas the quadrupole mode lies higher in energy (Supplementary Fig. 8), suggests, however, that the $\lambda_{corr} = 562$ nm mode remains capacitive in nature and is dominated by dipolar coupling. Furthermore, the structure in Fig. 4d with a continuous conductive Pd bridge between two gold NPs is conceptually similar to an all-Au dumbbell structure, for which the co-existence of localized coupled modes and long wavelength

standing modes as result of conductive coupling have been demonstrated before[13,74].

The determined gap conductance, $G$, provides a rational metric to characterize the evolution of the PR spectra as the concentration of Pd NPs in the gap and the interparticle separation increase from Fig. 4a–d. Moderate gap conductances of $G < 1 \times 10^{-1}$ S facilitated by individual Pd NP bridges, DNA mediated tunneling, or point-to-point tunneling along a network of Pd NPs decrease the charge accumulation on the two adjacent surfaces of otherwise strongly coupled Au NPs and induce a blue-shift of the capacitive BDP mode (Fig. 4a, b). The depolarizing effect of the gap conductance on the BDP resonance decreases as the charge accumulation decreases with increasing interparticle separation (Fig. 4c). However, once a sufficiently high number of Pd NPs is localized to the gap to fill it with touching Pd NPs, the gap conductance becomes high enough to induce the transition from capacitive to conductive coupling (Fig. 4d).

For completeness, Fig. 4e also includes an example of PRs without noticeable spectral shift ($\Delta\lambda = 0$), which comprises Au NP dimers with large separation ($S > 10$ nm). The interparticle separation of $S = 18$ nm is well above $S_{thresh}$ and the density of Pd NPs in the DNA is too low to establish a conductive contact between the Au NPs. Consequently, the observed plasmon resonance ($\lambda = 572$ nm, blue circles) is in good agreement with the simulated plasmon resonance ($\lambda = 573$, black line, orange line) for the PR.

**Removal of DNA from Pd NP containing PRs**. To further validate the relevance of DNA-bound Pd NPs for determining the spectral response of PRs, we measured the spectra of Pd NP containing, immobilized PRs with conductive (Fig. 5a) and capacitive (Fig. 5b) coupling before and after removal of DNA by incubation with an excess of mercaptoethanol. Difference plots are provided in Fig. 5c, d. For completeness, we also included the spectra of the same PRs before reduction of bound $Pd^{2+}$ to Pd NPs. Conductive coupling requires a high density of Pd NP in the gap region, establishing essentially a continuous bridge of NPs between the two Au NPs. These assemblies can be sufficiently stable so that replacement of DNA by the smaller and charge neutral mercaptoethanol does not remove the metal from the gap region but, instead, results in an even denser packing of Pd NPs and increased electron transport in an applied field. The associated gain in conductance is consistent with the strong red-shift of the delocalized mode and simultaneous elimination of the NP-localized plasmon mode (blue shoulder in original spectrum) in Fig. 5a. The PR dominated by capacitive coupling in Fig. 5b shows a blue-shift of the spectrum upon integration of Pd NPs. Based on our analysis from Fig. 4, we anticipate that this PR contains only a relatively low number of discrete NPs in the gap region. In this case, removal of the DNA matrix through mercaptoethanol removes the gap-localized Pd NPs, and the accompanying loss of gap conductance red-shifts the plasmon resonance back close to the initial value measured before $Pd^{2+}$ reduction. The experimentally observed spectral change observed upon DNA removal confirm the relevance of the DNA-Pd NP hybrid material in the gap region for determining the PR spectrum and reaffirm conductive and capacitive coupling models for the PRs in Fig. 5a, b, respectively.

**Discussion**

In this study, we have investigated the impact of interstitial Pd NP generation on the scattering spectra of DNA-connected dimers of Au NPs (PRs) in the visible range of the electromagnetic spectrum through correlation of single dimer spectroscopy, high-resolution TEM, and electromagnetic simulations. We have

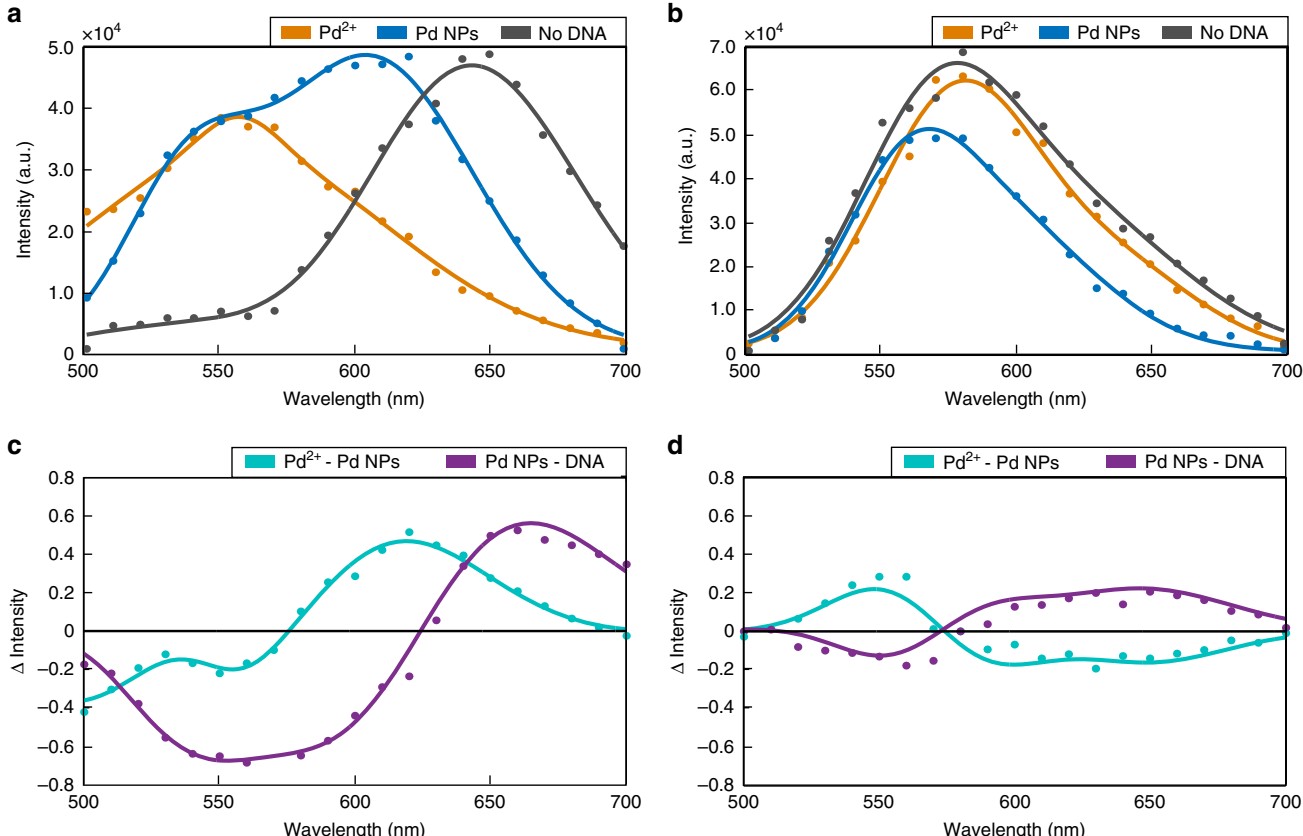

**Fig. 5** Effect of DNA removal on PR spectra. **a** Exemplary hyperspectral spectra collected from the same PR before reduction of $Pd^{2+}$ (orange), after reduction of $Pd^{2+}$ to Pd NPs (blue), and after DNA removal (gray). **b** Same data as in (**a**) but for a PR that shows a blue-shift upon Pd NP formation. **c** Difference spectra calculated from the normalized PR spectra for i.) $Pd^{2+}$ reduction to Pd NPs (blue) and ii) the removal of the DNA (purple) from (**a**). **d** Difference spectra calculated from the normalized PR spectra from (**b**)

shown that spectral effect of Pd NP generation depends sensitively on the gap width and the density of Pd NPs in the gap. The integration of interstitial Pd NPs increases the gap conductance and achieves a significant reduction of the charge pile-up on the opposite sides of the gap formed by strongly coupled Au NPs. This depolarization induces a measurable spectral blue-shift of the capacitive BDP resonance relative to the classical electromagnetic prediction. Intriguingly, we observed a blue-shift of the BDP in PRs containing discrete Pd NPs with a separation as large as 4.4 nm. A non-negligible gap conductance over this large separation is indicative of an efficient electron transport through subsequent electron tunneling between multiple NPs in the gap region. Higher concentrations of Pd NPs result in a partial filling of the gap with connected Pd NP and can establish singular contacts between the Au NP, which also result in a blue-shift of the BDP. Importantly, with increasing NP density in the gap, the gap conductance increases, which eventually induces the transition from capacitive to conductive coupling. This transition was indicated by a new peak in the long wavelength range of the PR spectrum.

DNA is known as a highly programmable and versatile linker, which facilitates a scalable fabrication of nanoplasmonic structures containing multiple coupled plasmonic components with pre-determined geometric arrangements by DNA-guided self-assembly. Our work shows that DNA is more than a simple structural element, but that its chemical modification provides opportunities for modulating the near- and far-field optical response of the assembled plasmonic nanostructures. Our findings suggest that it is feasible to change between different regimes of plasmon coupling in PRs by inserting varying amounts of Pd NPs into the gap region. The ability to detect and differentiate

different coupling regimes at a defined interparticle separation using the coupled plasmon as optical transducer has potential applications in nanoscale energy transfer, information processing, and sensing. A further refinement of the assembly strategies to enable networks of plasmonic NPs tethered by individual molecules, for instance with photo-switchable conductivities[75], will eventually result in experimental platforms that can greatly enhance active control of plasmons by fusing molecular electronics with nanoplasmonics.

## Methods

**Assembly of dsDNA/PEG functionalized Au NP dimers (PRs).** Aliquots of Au colloidal particles (41 ± 16 nm diameter) were mixed with two complimentary thiolated DNA handles (L-04: AAA TTG ATC TTA AAG TTG TTC AAC TAC CTA AAA AAG TTA A and L-05: T TAA CTT TTT TAG GTA GTT GAA CAA CTT TAA GAT CAA TTT) in ratios of 15:1 and 30:1 DNA/NP. After incubation, these DNA/NPs were combined with PEG-COOH (526 Da) to ensure the stability of the NP in solution. These DNA/PEG-NPs were washed thoroughly with deionized (DI) water and finally suspended in T80 buffer (10 mM Tris, 80 mM NaCl). The NPs were combined in equal concentrations and incubated in a 60 °C water bath for 24 + h before the samples were allowed to cool down to room temperature. The formed PRs were then isolated through gel electrophoresis. The dimer band were excised from the gel and immersed in DI water to recover the PRs. The average gap size of these dimers was 8 ( ± 7) nm. Following dialysis, the PRs were stored at 4 °C for up to a week.

**Generation of interstitial Pd NPs.** A solution of palladium acetate, $Pd(OAc)_2$, was made by dissolving 10 mg of palladium acetate in 1 mL DI water and sonicating for 10 min. The solution was centrifuged at 2500 g$^{-1}$ for 5 min to concentrate any undissolved palladium acetate. The supernatant was removed and diluted to 10 mL for use. An aliquot of the dsDNA/PEG Au NP dimers (200 μL) and the palladium acetate (200 μL) were combined and incubated while shaking for 4 h. This solution was washed through prepared 40 K desalting columns to remove the excess Pd

(OAc)$_2$, resulting in a solution of Pd$^{2+}$ containing dimers. Flow chambers or TEM grids were prepared by incubation with 0.1% poly-L-lysine for 15 min, followed by thorough washing with filtered DI H$_2$O. The Pd$^{2+}$-dimer solution from above was then incubated in the flow chamber or on the TEM grid for 5–15 min, depending on the concentration of the solution. After washing with filtered DI H$_2$O, these samples are incubated with DMAB solution (2.5 g/L sodium citrate, 2.5 g/L 85% lactic acid solution, 2.5 g/L DMAB) for 2 h. The samples were finally washed with filtered DI H$_2$O for a final time.

**Ratiometric imaging.** Pd$^{2+}$ loaded PRs were immobilized using poly-L-lysine (0.1%, 10 min of incubation in flow chamber) in a flow chamber and the reduction process was initiated by inducing a constant flow (0.8 µL/s) of the DMAB reducing agent. Optical measurements were performed on an Olympus IX71 inverted microscope under darkfield illumination (numerical aperture, N.A. = 1.2–1.4) using unpolarized whitelight from a 100 W Tungsten lamp. The scattered light was collected with a 60× oil objective (N.A. = 0.65) and then split using a dichroic mirror (580 nm) and filtered (550 nm & 600 nm, bandpass 10 nm). Due to different signal intensities, the green signal was enhanced during capture with an electron multiplier (EM) gain of 2. These signals were then collected at a rate of 1 frame/s with 2 back-illuminated CCD cameras (Andor iXon+) with a field of view of 75 × 75 µm$^2$, which were synced via a digital delay generator (Stanford Research Systems, DG645). Images were background corrected by subtracting the average of an area void of PRs and normalized to the starting area averaged intensity of each analyzed PR.

**Hyperspectral imaging.** Hyperspectral imaging (HSI) enables spectral analysis of ensembles through the use of a tunable filter and CCD camera. HSI was performed on another Olympus IX71 inverted microscope under darkfield illumination (N.A. = 1.2–1.4) using unpolarized whitelight from a 100 W Tungsten lamp. Scattered light from the sample was collected with a 60× oil objective (N.A. = 0.65) and imaged on a back-illuminated CCD (Andor iXon+). After an initial whitelight image was obtained, a tunable filter (CRi Varispec VIS, bandwidth = 10 nm) was placed in the path of the light and cycled from 500 nm to 700 nm with CCD images acquired every 10 nm. The experiments were performed at sufficiently low dilution so that the individual scatterers were well separated. All whitelight images were initially intensity filtered (the average monomer intensity plus one standard deviation was used as threshold), and for each emitter the integrated intensity over a 3 × 3 square pixel area was determined in each individual image. All intensities were background subtracted and corrected for the spectral profile of the incident light. The corrected intensities were fitted with a single Gaussian function or in cases where the spectra showed additional shoulder or other features with multiple Gaussian functions. Any signals without significant variance (Var < 0.1) of the signal intensities on the monitored wavelength channels (indicating a clear plasmon resonance) or with particularly poor fits ($R^2 < 0.7$) were excluded from the subsequent analysis. This process was repeated after the Pd$^{2+}$ reduction after application of a drift correction if necessary.

**Correlated darkfield and TEM single PR studies.** Samples prepared on TEM grids were first analyzed using darkfield spectroscopy to obtain scattering spectra and then inspected in the TEM. The TEM grids used had large windows (Ted Pella, Triple slot, Carbon Film) to allow for sufficient darkfield contrast. Patterns of marker beads, 2.0 µm polystyrene (PS), were used to ensure that the same dimers were analyzed in darkfield and TEM.

Pd$^{2+}$-loaded PRs were first electrostatically attached on a TEM grid and washed before a 4 mM DMAB solution was added for 120 min to reduce the DNA-bound Pd$^{2+}$. After removal of the DMAB and a subsequent washing step, darkfield analysis was performed on an inverted Olympus IX71 (N.A. = 1.2–1.4) microscope using unpolarized light from a 100 W Tungsten lamp. The scattered light was collected with a 60× oil objective (NA = 0.65), while the sample was immersed in glycerol and sandwiched between two slides. Spectra were recorded on an Andor Shamrock spectrometer, centered at 600 nm, and additional CCD images (Andor iXon+) were taken to determine the patterns of the PS beads and isolate the spectrally observed areas. After spectroscopy, the samples were cleaned through immersion in methanol to remove the glycerol. The samples were dried under a heat lamp and stored under vacuum until inspected in the TEM (FEI Tecnai Osiris).

**FDTD simulations.** Simulations for 40 nm Au dimers with 2–5 nm Pd NP(s) were carried out with Lumerical FDTD Solutions software (version 8.7), using the Johnson and Christy dielectric function for Au and the Palik dielectric function for Pd. Simulations assumed an ambient refractive index of $n_r = 1.57$ for glycerol to match the correlative darkfield/TEM experiments. The simulation area was a cube defined with 400 nm length on each size and a cubic mesh override region of 140 nm$^3$ with a 2 nm mesh size. An additional cubic override mesh with a 0.25 nm mesh size for the area immediately surrounding the NPs. A total field scattered field source was injected 120 nm below the plane of the dimers with a wavelength range of 300 to 900 nm and a polarization parallel to the dimer axis. The scattered light was detected with six 2D power detectors on each side of the mesh override region. The DNA conductivity was defined by a permittivity in the gap, following the method in ref. [20], with a conductivity of $\sigma_0 = 7.8$ S/m[64,65].

**Data availability.** All relevant data are available from authors upon request.

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

## Acknowledgements
This work was partially supported by the National Institutes of Health through grant 5R01CA138509 and the National Science Foundation through grant CHE-1609778.

## Author contributions
S.L. and B.M.R. conceived the idea and designed the experimental strategy. S.L. prepared the samples, performed all optical spectroscopy and TEM experiments, designed and analyzed all EM simulations, and analyzed all data. S.L. and B.M.R. interpreted the data, discussed the results, and wrote the manuscript.

## Additional information

**Competing interests:** The authors declare no competing interests.

