## [Peer Review File · Nature Communications]

Reviewers' comments:

Reviewer #1 (Remarks to the Author):

The work is interesting and well written and presented. Some points should be clarified before the publication:

- 1) in the introduction the motivation of the selection of Pd is missed, it should be clarified the selection of Pd
- 2) the same should be done to explain the final goal of interest of the present work
- 3) Before the figs. 4, to simulate by FTDT the experimental data, the authors introduces the gap conductance G , a better description of the adopted criterion to estimate G need to be introduced.

Reviewer #2 (Remarks to the Author):

Reviewer comments on the manuscript titled "Bridging the Gap: Effect of Interstitial Palladium on Plasmon-Driven Charge Transfer in Nanoparticle Dimers"

The authors investigate the effect of increasing Pd NP density inside the gap of two Au NPs separated by a distance. The optical response of the complexes are studied and the authors further claim that there is a capacitive to conductive plasmon coupling transition due to high conductance happening in the gap between Au NPs.

The text is well written. Most of the sections are easy to follow and supported by experimental measurements and at some point by simulation.

I would like to ask authors to address the following comments.

1. I do not see a clear practical application of this paper. In other words, what is the definite advantage of this research over the concurrent/previous works for charge/energy/data transfer between nanoparticles?
- 2- Numbers talk. Is it possible that the authors can give an explanation of how efficient their DNA assembled complexes bridge the gap between two Au NPs compared to similar works done in the field?
- 3- Figure 4d claimed to be an example of intensity spectrum showing both capacitance and conductance couplings. It is noteworthy to mention that at low energies, there might be multipolar radiation of closely packed NPs. How do the authors exclude the possibility of multipolar radiation and attribute the shoulder peak to only capacitance coupling?
- 4- It would be more illustrative if the light polarization is also shown on Figure 4 in the right column.

Reviewer #3 (Remarks to the Author):

The manuscript "Bridging the Gap: Effect of Interstitial Palladium on Plasmon-Driven Charge Transfer

in Nanoparticle Dimers" by Lerch & Reinhard investigates the effect of DNA-templated Palladium growth in the gap between two gold nanoparticles.

Darkfield scattering spectroscopy of individual particle dimers reveals pronounced shifts of the plasmon resonances after Pd particle growth. Shifts to both shorter (bluer) and higher (red) wavelength are observed. The authors use correlated TEM imaging (images of individual particle dimers are correlated with the respective spectra) to obtain additional information such as particle-particle distance and Pd particle density in the gaps.

The authors conclude, that a combination of DNA-mediated charge transport and gap conductance based on tunnelling events via the Pd particles is responsible for the observed blue-shifted spectra. By tweaking the conductivity inside the gap space, FDTD simulations are tuned to match the experimental data. Generally, I think the authors show an interesting effect – the possibility of gap conductance – that has not yet received much coverage in the literature on plasmonic hot spots. I could agree with the interpretations of the authors, but the experimental evidence is a little thin. My main concern is, that blue-shifted spectra in DNA-assembled particle dimers are usually not reported by other groups, in fact, almost all studies solely report red shifted plasmon resonance spectra for particle dimers due to the distance dependent plasmon coupling. This holds true also for gap sizes below the here given threshold of 2.7nm. The cited references showing that DNA-mediated gap conductivity impacts the plasmon resonance are by the authors of the here presented manuscript. Additionally, the sentence "The dimer band were excised from the gel and immersed in DI water to recover the PRs." (first paragraph in the experimental section) made me wonder about the actual nature of the connection between the gold particles to form dimers. DNA double strands usually disassociate in the absence of salts that screen the strongly charged DNA backbone. Therefore I could imagine that many or some of the dimers are coupled by unspecific interactions, such as direct Au surface-to-surface aggregation. Conductance between particles in such dimers are not surprising. The authors would need to show many more spectra with corresponding TEM images to convince me here.

Another aspect of the assembly that I do not understand is the following: how can a small variation (factor of two) in DNA excess over particles (30:1 vs. 15:1) have such a pronounced effect on the DNA density around the particles? Also here I would have hoped to see a wider range of densities being studied (not necessarily by spectroscopy, but by gel and TEM). Generally, gel analysis (e.g. in the SI) would help to access the quality of the assemblies.

Along these lines, the authors claim on the 4th page of the text that "a typical field of view" contains 400 scatterers. The darkfield images in Figs. 2 & 3 do not support this claim. The SI generally shows little additional data and should be expanded by such images and many more TEM images correlated to spectra.

In conclusion I do not recommend publication of this work in its present form. The authors should discuss possible controversies concerning the origin of the observed blue shifts and rationalize their simulations that include the gap conductances more thoroughly.

Minor comments:

- abstract, line 16: "with γ_a rate constant ..."
- line 20: "rate constant rate" ?
- last page, line 47ff: What do the numbers indicate? Apparently "opening probabilities", but within which time frame?
- Some references related to single-molecule FRET on strand displacement reactions (e.g. Lamb&Simmel, Nano Letters) and to strand displacement cascades (some works of Georg Seelig and or Peng Yin) are missing.

Reviewer #1 (Remarks to the Author):

Comment 1: The work is interesting and well written and presented. Some points should be clarified before the publication:

Response: We thank the reviewer for the overall positive evaluation.

Comment 2: 1) in the introduction the motivation of the selection of Pd is missed, it should be clarified the selection of Pd

Response: We added on page 5, line 3-5: "*Pd was chosen as Pd²⁺ cations are known to bind to DNA, and hierarchical Pd – Au heterostructures have interesting sensor applications*"

Comment 3: 2) the same should be done to explain the final goal of interest of the present work

Response: In response to the reviewer's question we clarified on page 5, line 5-9: "*We characterize the effect of discrete Pd NPs localized in the gap on BDP depolarization in the capacitive coupling regime through correlated optical single dimer spectroscopy and high-resolution transmission electron microscopy (TEM) and test the hypothesis that the generation of Pd NP in the gap of DNA-connected plasmonic molecules provides a chemical strategy to tune their collective plasmon resonance after self-assembly*⁵⁹"

We also added on page 17, line 18-27: "*DNA is known as a highly programmable and versatile linker, which facilitates a scalable fabrication of nanoplasmonic structures containing multiple coupled plasmonic components with pre-determined geometric arrangements by DNA-guided self-assembly. Our work shows that DNA is more than a simple structural element, but that its chemical modification provides opportunities for modulating the near- and far-field optical response of the assembled plasmonic nanostructures. Our findings suggest that it is feasible to change between different regimes of plasmon coupling in PRs by inserting varying amounts of Pd NPs into the gap region. The ability to detect and differentiate different coupling regimes at a defined interparticle separation using the coupled plasmon as optical transducer has potential applications in nanoscale energy transfer, information processing, and sensing*"

Comment 4: 3) Before the figs. 4, to simulate by FDTD the experimental data, the authors introduces the gap conductance G, a better description of the adopted criterion to estimate G need to be introduced.

Response: In response to the reviewer's suggestion, we added the following section that introduces G before our discussion of Figure 4 on page 11, line 17-19: "*We also added the effective gap conductance, G, of a homogenous medium filling the space between the two Au NPs that in FDTD simulations best reproduces the experimental spectra of the individual PR in the right column of Figure 4.*"

Reviewer #2 (Remarks to the Author):

Comment 1: Reviewer comments on the manuscript titled "Bridging the Gap: Effect of Interstitial Palladium on Plasmon-Driven Charge Transfer in Nanoparticle Dimers"

The authors investigate the effect of increasing Pd NP density inside the gap of two Au NPs separated by a distance. The optical response of the complexes are studied and the authors further claim that there is a capacitive to conductive plasmon coupling transition due to high conductance happening in the gap between Au NPs.

The text is well written. Most of the sections are easy to follow and supported by experimental measurements and at some point by simulation.

Response: We thank the reviewer for the overall positive evaluation of the manuscript.

Comment 2: I would like to ask authors to address the following comments.

1. I do not see a clear practical application of this paper. In other words, what is the definite advantage of this research over the concurrent/previous works for charge/energy/data transfer between nanoparticles?

Response: We agree with the reviewer that there currently a lot of interest in nanoplasmonic assemblies to connect photonics with electronics and to facilitate coherent transfer of energy and information between nanoparticles and that researchers pursue many different approaches. One common fundamental challenge in realizing these approaches is, however, the need for a scalable fabrication. For plasmonic nanostructures, DNA guided self-assembly (including origami and other approaches) is considered a viable pathway to a scalable fabrication of a complex nanocircuitry due to the unique four base code of DNA that makes the realization of even complex structures feasible. Our work is important as it demonstrates that DNA is not necessarily an inert structural element in these structures. Furthermore, we show that by integration of small metal nanocrystals into the DNA scaffold the physical properties of the plasmonic assembly can be modulated, which provides new opportunities for switchable plasmonic nanostructures.

To address the reviewer's comment, we added on 17, line 18-27: "DNA is known as a highly programmable and versatile linker, which facilitates a scalable fabrication of nanoplasmonic structures containing multiple coupled plasmonic components with pre-determined geometric arrangements by DNA-guided self-assembly. Our work shows that DNA is more than a simple structural element, but that its chemical modification provides opportunities for modulating the near- and far-field optical response of the assembled plasmonic nanostructures. Our findings suggest that it is feasible to change between different regimes of plasmon coupling in PRs by inserting varying amounts of Pd NPs into the gap region. The ability to detect and differentiate different coupling regimes at a defined interparticle separation using the coupled plasmon as optical transducer has potential applications in nanoscale energy transfer, information processing, and sensing"

Comment 3: 2- Numbers talk. Is it possible that the authors can give an explanation of how efficient their DNA assembled complexes bridge the gap between two Au NPs compared to similar works done in the field?

Response: In response to the reviewer's concern, we have included gels of the assembled plasmon rulers in the new Supporting Information Figure 1. We used these gels to determine approximate yields of the assembled dimers by analysis of the relative intensities of the product bands. According to the gels, the yield of the plasmon ruler is approximately 20%. This information is included in legend of the new Supporting Information Figure 1. A detailed comparison with other studies is challenging and goes beyond the scope of this work as the number of DNA strands in the synthesis and the length of the DNA linkers varies between different studies.

Comment 4: 3- Figure 4d claimed to be an example of intensity spectrum showing both capacitance and conductance couplings. It is noteworthy to mention that at low energies, there might be multipolar radiation of closely packed NPs. How do the authors exclude the possibility of multipolar radiation and attribute the shoulder peak to only capacitance coupling?

Response: We agree with the reviewer that multipolar modes can couple to far-field radiation in nanoparticle assemblies. The multipolar modes in nearly touching dimers have, for instance, been investigated by Atay and coworkers [see ref. 71 in our revised manuscript]. These higher order angular modes were, however, still blue relative the monomer resonance (see also the new Supporting Information Figure 8). The modes of interest in this work lie higher in energy than the classical electromagnetic prediction for dimers of the same interparticle separation, but they remain energetically lower than the monomers. Furthermore, the charge distribution plot for the 562 nm mode in Figure 4d has still strong similarities with a dipolar mode. A last argument in favor of our assignment is that this particular configuration is structurally similar to that of a dumbbell structure for which the coexistence of dipolar and long wavelength standing mode has been shown before (see ref. 69 in the revised manuscript).

To address the reviewer's concern, we added on page 14, line 18 – page 15, line 2: *"The mode at $\lambda_{corr} = 562$ nm could, in principle, represent a quadrupole mode, as these have been shown to couple to far-field radiation in dimers of touching NPs.⁷¹ The simulated charge distribution and the fact that the mode remains lower in energy than the BDP of an uncoupled monomer, whereas the quadrupole mode lies higher in energy (Supporting Information Figure 8), suggests, however, that the $\lambda_{corr} = 562$ nm mode remains capacitive in nature and is dominated by a dipolar coupling. Furthermore, the structure in **d** with a continuous conductive Pd bridge between two gold NPs is conceptually similar to an all Au dumbbell structure, for which the co-existence of localized coupled modes and long wavelength standing modes as result of conductive coupling have been demonstrated before.^{12,69}"*

Comment 5: 4- It would be more illustrative if the light polarization is also shown on Figure 4 in the right column.

Response: Following the reviewer's suggestion we indicated the light polarization in the right column of Figure 4, first row.

Reviewer #3 (Remarks to the Author):

Comment 1: The manuscript "Bridging the Gap: Effect of Interstitial Palladium on Plasmon-Driven Charge Transfer in Nanoparticle Dimers" by Lerch & Reinhard investigates the effect of DNA-templated Palladium growth in the gap between two gold nanoparticles.

Darkfield scattering spectroscopy of individual particle dimers reveals pronounced shifts of the plasmon resonances after Pd particle growth. Shifts to both shorter (bluer) and higher (red) wavelength are observed. The authors use correlated TEM imaging (images of individual particle dimers are correlated with the respective spectra) to obtain additional information such as particle-particle distance and Pd particle density in the gaps.

The authors conclude, that a combination of DNA-mediated charge transport and gap conductance based on tunnelling events via the Pd particles is responsible for the observed blue-shifted spectra. By tweaking the conductivity inside the gap space, FDTD simulations are tuned to match the experimental data. Generally, I think the authors show an interesting effect – the possibility of gap conductance – that has not yet received much coverage in the literature on plasmonic hot spots.

Response: We thank the reviewer for stressing the general relevance of our work.

Comment 2: I could agree with the interpretations of the authors, but the experimental evidence is a little thin. My main concern is, that blue-shifted spectra in DNA-assembled particle dimers are usually not reported by other groups, in fact, almost all studies solely report red shifted plasmon resonance spectra for particle dimers due to the distance dependent plasmon coupling. This holds true also for gap sizes below the here given threshold of 2.7nm. The cited references showing that DNA-mediated gap conductivity impacts the plasmon resonance are by the authors of the here presented manuscript.

Response: We do not question that the resonances in DNA-connected nanoparticles are red-shifted when compared to that of the individual particles. Our "blue-shifts" are relative to the electromagnetic prediction and indicate that the measured shift is weaker than predicted by classical electrostatic coupling models. To improve the experimental evidence of this relative blue-shift, we have included additional data in Supporting Information Figures 5 and 6 that show spectra and correlated structures of 20 plasmon rulers with different interparticle separations and Pd NP loadings. The additional data support our conclusion that DNA can induce a spectral blue-shift relative to the classical electromagnetic prediction at short separations and that the range of the blue-shift is extended if Pd NP are incorporated into the DNA.

Regarding the reviewer's comment that most studies report a red-shift even at very short separations, we would like to point out that our system differs in important aspects from other studies. To the best of our knowledge, our work is the only work that has performed a correlation of high resolution structural electron microscopy and single dimer optical spectroscopy for DNA-linked NPs to independently measure both resonance wavelength and interparticle separation with high resolution in the few nm separation range. This experimental approach requires specific experimental conditions. For instance, conventional plasmon ruler measurements are usually performed in aqueous solution while our experimental spectra are measured after drying and immersion in glycerol or oil. Other DNA-based structures with short interparticle separations have been measured in the dried state. Sub-5 nm gaps have, for instance, been realized by Baumberg and coworkers (see ref 53 in our manuscript) based on DNA origami structures. But different than in our experimental approach where DNA connects the NP and is located right between the NPs, the origami approach generated binding grooves on which the NP were bound. The interparticle gap volume in these structures was only partially filled with DNA strands, whose helices were furthermore oriented orthogonal to the interparticle axis. All of these differences can lead to differences in the coupling mechanism.

To clarify these points and to address the reviewer's concern related to the observation of a blue-shift in our previous work, we added a critical discussion of our finding in the context of studies from other groups on page 4, line 6-21: *"The role of DNA-mediated conductance in the transition from the classical electromagnetic to quantum mechanical coupling regime in DNA-connected NPs raises important fundamental questions. Conventional electromagnetic coupling, which predicts a continuous red-shift with decreasing gap separation, provides in general an adequate description of practical DNA-linked NP assemblies in solution, but correlated optical and high resolution characterization on the single dimer level have provided indications that at short separations the DNA itself influences the coupling between gold (Au) and silver (Ag) NP dimers.^{29,42} Significant blue-shifts of the BDP mode relative to the classical electromagnetic prediction for distances of up to S_{thresh} =*

2.7 nm observed for immobilized dimers immersed in oil or glycerol were attributed to DNA-mediated charge transfer between the gap defining NPs. Similar blue-shifts were observed – albeit at shorter separations – with other molecular linkers that provide gap conductance.^{27,28,33} Intriguingly, DNA origami templated dimers with sub-5 nm interparticle separations, whose gap region was only partially filled with DNA strands oriented perpendicular to the interparticle axis,⁵³ did not show signs of a weakened electromagnetic coupling. Together, these observations indicate a sensitive dependence of the optical response of self-assembled NP dimers with short interparticle separations on details of the gap configuration.”

A last point we would like to mention is that other groups have shown similar relative spectral blue-shifts with other linker molecules that increased the gap separation over which tunneling can occur via through-bond-tunneling. We refer here to work, for instance, by Nijhuis and coworkers (ref. 15, 28). While these other studies were not performed with DNA, there is independent evidence that DNA can provide delocalized conductive states over separations up to 10 base pairs for hydrated and partially dried DNA. Please refer here, in particular, to studies by Tan et al. (ref. 52 in our manuscript) or Barton and coworkers (ref. 51 in our work). These previous studies are consistent with our findings and are cited and discussed in our manuscript.

Comment 4: Additionally, the sentence "The dimer band were excised from the gel and immersed in DI water to recover the PRs." (first paragraph in the experimental section) made me wonder about the actual nature of the connection between the gold particles to form dimers. DNA double strands usually disassociate in the absence of salts that screen the strongly charged DNA backbone.

Response: We agree with the reviewer that DNA can de-hybridize in DI water. However, in our experiment – due to the salt contained in the agarose gel and Tris-borate (TBE) running buffer – the dimer containing solution still contains substantial concentrations of salt. It has been shown that tris cations stabilize the negatively charged phosphate backbone in a similar fashion as NaCl [Stellwagen et al. Biopolymers 54, 137 (2000)]. To address the reviewer’s comment we measured the salt concentration in the supernatant. We found that it still contains approximately 20 mM TBE.

Comment 5: Therefore I could imagine that many or some of the dimers are coupled by unspecific interactions, such as direct Au surface-to-surface aggregation. Conductance between particles in such dimers are not surprising. The authors would need to show many more spectra with corresponding TEM images to convince me here.

Response: In response to the reviewer’s comment we have added additional correlated TEM images / darkfield spectra in Supporting Information Figures 5 and 6. These additional spectra and corresponding images confirm well separated Au nanoparticles. It is clear that with increasing Pd nanoparticle concentration, the Pd nanoparticles in the gap can establish conductive “bridges” between the gold nanoparticles. This effect and its impact on the measured spectra is discussed on pages 13-15.

We also point out that the controls (dimers without Pd²⁺) in Figure 3b underwent the same assembly and purification but did not show any significant spectral shifts after treatment with reducing agent. If unspecific interactions, such as direct Au surface-to-surface contacts were responsible, these should be detected in the controls. The absence of any systematic blue-shifts in the controls argues against non-specific contacts between Au nanoparticles as cause for the observed blue-shift.

Comment 6: Another aspect of the assembly that I do not understand is the following: how can a small variation (factor of two) in DNA excess over particles (30:1 vs. 15:1) have such a pronounced effect on the DNA density around the particles? Also here I would have hoped to see a wider range of densities being studied (not necessarily by spectroscopy, but by gel and TEM). Generally, gel analysis (e.g. in the SI) would help to access the quality of the assemblies.

Response: The number of Pd NP bound to the different experimental conditions shows some natural spread. For all experimental conditions, we observed gold dimers with a high or low number of Pd NPs, but the contribution from high Pd NP loadings increased with DNA surface concentration. To make this point clearer and to address the reviewer’s concerns, we quantified the contribution from Au dimers with low and high Pd NP loading for three different DNA:NP ratios (15:1, 22:1, and 30:1). The contribution from the high Pd NP configuration increases from 40.1% over 49.8% to 67.7%. This information was added as Table 2 in the Supporting Information of the revised manuscript. We added in the text on page 6, line 17-19: “For PRs, the

number of Pd NP showed some variability, but the number and average size of Pd NPs bound to the DNA-functionalized NP increased overall with DNA loading (Supporting Information Table 2)."

In response to the reviewer's comment about the analysis of the dimers in the SI, we have provided a more detailed characterization of the dimer formation in the Supporting Information. We included gels for the assembled plasmon ruler in the Supporting Information Figure 1. The data confirm the successful formation of DNA-linked NP dimers.

Comment 7: Along these lines, the authors claim on the 4th page of the text that "a typical field of view" contains 400 scatterers. The darkfield images in Figs. 2 & 3 do not support this claim. The SI generally shows little additional data and should be expanded by such images and many more TEM images correlated to spectra.

Response: The images shown in the manuscript were only extracts from much larger field of views. We showed magnified views of smaller regions to improve the visibility of the individual particles. In response to the reviewer's comment, we have included an image for an entire field of view in Supporting Information Figure 2. We have also provided additional TEM images and correlated spectra in Supporting Information Figures 5 and 6. The large number of plasmon ruler in the field of view allows the collection of large number of spectral shifts data as shown in Figure 3. But for the correlated TEM/optical spectroscopy experiments the throughput is limited by the correlation of TEM and optical spectroscopy which have vastly different levels of magnifications. We have made the necessary changes in the manuscript and figure legends to clarify that the images in Figure 3 and 4 are only sub-sections of a larger field of view.

Comment 8: In conclusion I do not recommend publication of this work in its present form. The authors should discuss possible controversies concerning the origin of the observed blue shifts and rationalize their simulations that include the gap conductances more thoroughly.

Response: In response to the reviewer's constructive feedback, we have revised our manuscript to address the raised concerns. In particular, we have added additional data that support the existence of a spectral blue-shift as discussed in our manuscript (see also response to Comment 5) and we have discussed the spectral blue-shift for DNA-connected NPs in the context of other studies of similar systems (see response to Comment 8) as suggested by the reviewer. Throughout the manuscript we have elaborated on the performed simulations that provide a model to account for the observed spectral behavior based on changes in the gap conductance.

Comment 9:

Minor comments:

- abstract, line 16: "with a rate constant ..."
- line 20: "rate constant rate" ?
- last page, line 47ff: What do the numbers indicate? Apparently "opening probabilities", but within which time frame?
- Some references related to single-molecule FRET on strand displacement reactions (e.g. Lamb&Simmel, Nano Letters) and to strand displacement cascades (some works of Georg Seelig and or Peng Yin) are missing.

Response: These comments seem to be included by mistake by the reviewer as they do not relate to our manuscript. We ignored these points in the preparation of our revision.

REVIEWERS' COMMENTS:

Reviewer #2 (Remarks to the Author):

2nd round of review of the manuscript "Bridging the Gap: Effect of Interstitial Palladium on Plasmon-Driven Charge Transfer in Nanoparticle Dimers":

I believe that the authors completely addressed my comments/concerns in the revised version of the manuscript.

The paper now is well-justified regarding my comments and it is scholarly written.

On my part, the manuscript is publishable.

Regards.

Reviewer #3 (Remarks to the Author):

The authors addressed all comments by the referees and thus improved the manuscript considerably. It is now ready for publication.